# Manufacturing and Joining PP/NBR Blends in the Presence of Dual Compatibilizer and Halloysite Nanotubes

**DOI:** 10.3390/nano13010049

**Published:** 2022-12-22

**Authors:** Sina Alipour, Amir Mostafapour, Hossein Laieghi, Anna Marzec

**Affiliations:** 1Faculty of Mechanical Engineering, University of Tabriz, Tabriz P.O. Box 51666-16471, Iran; 2Department of Mechanical Engineering, Gazi University, Ankara 06570, Türkiye; 3Additive Manufacturing Technologies Research and Application Center-EKTAM, Gazi University, Ankara 06560, Türkiye; 4Institute of Polymer and Dye Technology, Faculty of Chemistry, Lodz University of Technology, 90-924 Lodz, Poland

**Keywords:** polypropylene (PP), halloysite nanotubes (HNTs), nanocomposite, friction stir welding (FSW), mechanical properties, microstructural evaluation

## Abstract

Polypropylene (PP)/acrylonitrile butadiene rubber (NBR) composite plates reinforced with halloysite nanotubes (HNTs) were manufactured in the presence of dual compatibilizers: PP-grafted maleic anhydride (PP-g-MA) and styrene ethylene butylene styrene-grafted maleic anhydride (SEBS-g-MA). The mechanical characteristics and microstructure of the PP/NBR/HNT nanocomposites were investigated as a function of NBR content (10, 20, and 30 wt.%) and HNTs content (3, 5, and 7 wt.%). The results demonstrated that the rubber particles were well dispersed over the PP matrix and that the HNTs were partly agglomerated at contents above 5%. Friction stir welding (FSW) was used to join the nanocomposite plates. A significant reduction in scattered NBR droplet size was seen in the FS-welded specimens containing 80/20 (wt/wt) PP/NBR composites in the presence of a dual compatibilizer. Considerable improvement in particle dispersion was observed in the case of PP/NBR blends filled 80/20 (wt/wt) with HNTs joined using FSW, leading to enhanced mechanical properties in the joints. This was due to the stirring action of the FSW tool. Suitable agreement between anticipated and confirmed values was observed in experiments.

## 1. Introduction

The characteristics of polymer matrices can be modified by reinforcing them with nanoparticles or blending them with other polymers [1,2,3]. Ultrafine inorganic particles have increasingly been employed to enhance the overall performance of thermoplastic materials. The final characteristics of the nanocomposite materials are significantly influenced by the dispersion and interfacial bonding of inorganic particles in the matrix [4]. Thermoplastic nanocomposites containing inorganic nano-fillers have attracted considerable attention from researchers and industry owing to their unique characteristics and intrinsic advantages, such as strong resistance to impact, relatively low-cost storage, recyclability, and weldability [5]. Polyethylene (PE), polyamide (PA), polystyrene (PS), polycarbonate (PC), and polypropylene (PP) are some of the most common materials used as matrices in the fabrication of thermoplastic nanocomposites [6]. By blending two or more polymers, the advantages of the second polymer can counteract the drawbacks of the first. The characteristics of ternary hybrid systems can be effectively monitored via the properties of phases, the morphology of the nano-filler, the interfacial interaction of constituents, etc.

Due to its well-balanced mechanical and physical characteristics, as well as easy processability, PP has become a widely used thermoplastic material. However, PP has poor toughness as a result of its high glass transition temperature (Tg), which restricts its applicability in various areas. Therefore, the fabrication of a ternary hybrid system comprised of PP as a matrix, a nonreactive polar elastomer including acrylonitrile butadiene rubber (NBR) and ultrafine inorganic particles as minor phases appears to be a rational approach to broadening its range of applications. However, compatibilization with low-polarity polymers such as PP is often challenging owing to the large polarity discrepancy between phases. Studies on the mechanical and morphological characteristics of PP/NBR composites have demonstrated that incorporating maleic-modified PP and phenolic-modified PP as compatibilizing agents in PP/NBR blends may result in a suitable balance between toughness and stiffness by increasing interfacial bonding between phases [7,8,9].

Several studies have been conducted on the compatibilization of polymer blends using single or dual compatibilizers. Refined morphology and consequent improvement of the characteristics of the PP-based composites have been achieved by using a single compatibilizer in the blend. For example, PP grafted with acrylic acid was employed as a compatibilizer in PP/ABS blends. The PPg-acrylic acid improved the shape and mechanical characteristics of the blend [10]. Using a dual compatibilizer, on the other hand, can improve the molecular structure of PP/ABS composites [11]. The electrical conductivity of MWNT-filled PP/ABS blends and their composites increase when dual compatibilizers are used [12]. Using a dual compatibilizer in poly(ethylene terephthalate) (PET) and polystyrene (PS) blends results in a smaller phase domain, increased viscosity, and enhanced mechanical characteristics [13]. Similar outcomes are seen in PP and poly(butylene terephthalate) (PBT) blends [14]. The effect of a dual compatibilizer on the morphological and mechanical characteristics of PP/ABS blends was analyzed. The observed refinement in morphology led to an improvement in their mechanical characteristics. PP-g-MA and styrene ethylene butylene–styrene triblock copolymer grafted with maleic anhydride (SEBS-g-MA) were used as compatibilizers. PP/ABS blends with and without PP-g-MA and SEBS-g-MA as dual compatibilizers were compared. It was found that dual compatibilizers are far more effective than single compatibilizers at improving the morphological and mechanical characteristics of the blends [15].

The incorporation of an NBR phase into a PP matrix has the potential to enhance the toughness of the blend. However, Introducing NBR into a PP matrix deteriorates its mechanical properties, such as tensile strength and modulus. To achieve an optimum combination of strength and toughness, a small amount of nano-filler can be added to the PP/NBR blends. While montmorillonite (MMT) and silica have long been employed as naturally occurring inorganic nano-fillers, there is increasing interest in the use of nano-fillers with high aspect ratios, such as halloysite nanotubes (HNT). HNT-based polymer nanocomposites have been the subject of numerous investigations in recent years [16,17,18,19]. They offer an environmentally friendly, novel, and promising reinforcement material for thermoplastics [20]. In comparison to other nano-fillers, HNTs are more easily disseminated in a polymer matrix due to their comparatively low hydroxyl density on exterior surfaces and restricted intertubular contact area. HNTs with geometry similar to that of carbon nanotubes (CNTs) are easily accessible and considerably cheaper than other nano-fillers. Polymers such as polypropylene (PP) can benefit from the presence of HNTs, which promote nucleation and modification of polymer crystal structures [21,22,23]. Due to their hollow shapes and high aspect ratio, HNTs can be used for the fabrication of low-density polymer nanocomposites with significantly improved mechanical properties [24]. Consequently, HNTs are considered appropriate nano-fillers for the production of polymer nanocomposites.

To effectively use thermoplastic nanocomposites as a substitute for large conventional structures, it is essential to use appropriate joining methods to assemble the plates. Heat sources are critical to the efficiency of the process, the final quality of components, and processing costs [25,26]. Friction stir welding (FSW) has significant advantages over other joining procedures since it avoids the production of porosity, solidification cracking, and residual thermal stress [27,28,29,30]. Tool-to-workpiece friction generates heat, so no melting of the workpieces is required in this method. Rotation and pressure imparted to the weld line by the tool are used to first soften, then mix the materials [31]. Recently, a heating-assisted FSW approach to joining PA6/NBR/HNT nanocomposite plates was described [32,33].

In the present study, a polymer matrix nanocomposite (PP/NBR) with two types of compatibilizer (PP-g-MA and SEBS-g-MA) was prepared using a twin screw extruder. The influence of HNTs on the microstructural and mechanical properties of the PP/NBR blends was investigated. The heating-assisted FSW technique was evaluated for joining the PP/NBR/HNTs nanocomposite plates. The dual compatibilized PP/NBR/HNTs nanocomposites and FS-welded plates were characterized utilizing scanning electron microscopy (SEM) with energy dispersive X-ray analysis (EDX), as well as by tensile, hardness, and impact testing. The mechanical and microstructural performances of the FS-welded samples were compared with extruded PP/NBR/HNTs blends.

## 2. Materials and Methods

### 2.1. Materials and Composite Preparation

Polypropylene (HP502N) with a density of 0.9 g/cm^3^ and a melt flow index of 12 g/10 min at 230 °C was supplied by Arak Petrochemical Company (Arak, Iran). Nitrile rubber (KNB-35L) containing 34 wt.% of acrylonitrile was purchased from Kumho (Seoul, Republic of Korea). HNTs with a density of 2.55 g/cm^3^ and brightness of 98.9% were supplied by Imerys Tableware, Asia Limited, Auckland, New Zealand. Polypropylene-grafted maleic anhydride (PP-g-MA) with 1 wt.% of maleic anhydride content, density of 0.9 g/cm^3^, and MFI of 10 g/10 min was purchased from Aria Polymer Company (Isfahan, Iran). Styrene ethylene butylene styrene triblock copolymer grafted with maleic anhydrite (SEBS-g-MA) with a melt flow index of 14 g/10 min was purchased from Karton, India.

Melt blending of the HNT powder and the PP granules in the presence of a dual compatibilizer was performed using a Brabender co-rotating TSE (ZSK, Krefeld, Germany). Subsequently, 20 wt.% of NBR was added. The temperature profile in the extruder from the hopper to the die was set at 195–230 °C for six minutes at a screw speed of 80 rpm. The prepared granules were molded using a hot press at 260 °C for 5 min. The resulting sheets of PP/NBR/HNT nanocomposites were cut using a miter saw to dimensions of 80 × 40 × 5 mm for joining by heat-assisted FSW.

### 2.2. Experimental Procedure

A tool comprising a stationary shoe-shaped shoulder, a rotating pin, and a heater located inside the shoulder was used in the research. The tool pin and shoulder were made of H13 hot-working steel owing to its high-temperature resistance and mechanical strength. After machining the stationary shoe-shaped shoulder to dimensions of 30 × 160 mm, a polytetrafluoroethylene (PTFE) coating was applied. Three tool pin geometries were selected, as shown in Figure 1a: (1) a threaded straight cylindrical pin; (2) a threaded tapper cylindrical pin; and (3) a threaded square pin. The prepared sheets were positioned on the backing plate to keep the workpieces stable throughout the FSW procedure depicted in Figure 1b. Preliminary research showed that the fabricated PP/NBR/HNT sheets could only be welded within a limited range of process parameters (Table 1).

### 2.3. Characterization

An ASTM D638 (type V) [34] was used to prepare dog bone-shaped samples of the nanocomposites. Tensile testing was performed using a GOTECHAI-7000 universal testing machine (GOTECH Testing Machines Inc., Taichung City, Taiwan) with a 5 mm/min crosshead speed. To guarantee the validity of the test, at least five replicates were conducted under similar conditions. Izod impact tests based on ASTMD256 [35] were performed using an SIT-20D machine (ZwickRoell Ltd., Worcester, UK). The specimens used for the Izod impact test were notched using a V-Notch sampling machine model GT-7016-A2 (GOTECH Testing Machines Inc., Taichung City, Taiwan). Each specimen was analyzed in five replicates, and the average value was calculated. Following the ASTMD2240 (type D) [36] test method and using a Shore D (SD) hardness machine (ZwickRoell Ltd., Worcester, UK), the hardness behavior of the welded samples was measured. A cross-section of the nugget zone was measured to determine the hardness. Scanning electron microscopy (SEM) was performed to study the morphologies of the specimens. The fractured tensile samples were etched for 24 h to remove the NBR phase from the samples. Then, gold coatings were applied to the etched samples. The SEM analysis was performed using a Tescan SEM model MIRA3 FEG (TESCAN, Kohoutovice, Czech Republic).

## 3. Results and Discussion

In this study, three different concentrations of NBR (10, 20, 30 wt.%) were added to PP with 10 wt.% of a dual compatibilizer consisting of SEBS-g-MA and PP-g-MA. The mechanical characteristics of the PP/NBR composite materials are listed in Table 2. As a result of the addition of a dual compatibilizer, PP becomes polarized and more compatible with NBR. Due to the dipolar interaction between the MA group of a dual compatibilizer and NBR, the interfacial tension is reduced, and the dipolar interaction between PP and NBR is increased. The distribution of the NBR droplets across the PP matrix is shown in Figure 2. The rubber droplets that were removed by etching in acetone are visible as dark holes. As can be seen by comparing Figure 2a,c, larger concentrations of NBR resulted in poorer particle dispersion compared to low concentrations, and the rubber particles in the sample with 30 wt.% NBR are dispersed in a nonsymmetric way. Thus, the sample containing 20 wt.% of NBR has the most acceptable characteristics (see Figure 2b).

Joining thermoplastic composites is an essential step in the fabrication of larger and more sophisticated parts. In this study, we proved the feasibility of FS welding polypropylene-based composites using a rotational speed of 560–900–1120 rpm and a traverse speed of 14–28–40 mm/min. Based on previous studies in the literature [37,38,39,40,41,42], a threaded cylindrical pin profile was selected as the best type of pin profile to obtain the parametric range needed for the FSW process.

The process parameter ranges were initially chosen based on a visual evaluation of weld quality. Since polymeric materials react differently at various temperatures, it became important to find the optimum parameters for making sound joints. The PP/NBR composites were found to have acceptable quality within a given parametric range, but if FSW was performed outside that range, it would not result in the requisite quality.

At rotational speeds higher than 1120 rpm, the excessive heat generated destroys the polymer chains and darkens the polymer weld zone. Mixing also does not work well because of the low heat generated at rotational speeds below 560 rpm. Consequently, instead of melting and mixing, the tool pin pierces the inside of the part, preventing joining from taking place. When the heat generated by both the rotational speed and the shoe is high, mixing does not occur properly, and molten materials move out of the shoulder owing to their greater fluidity. Additionally, the degradation of the polymer chains can be accelerated at higher temperatures. In the case of traverse speed, the excessive stirring occurs at speeds lower than 14 mm/min. Increasing the duration of the tool pin rotating in a specific area of the sample allows heat to build up in that area, degrading the polymer. When the traverse speed exceeds 40 mm/min, the tool pin is unable to fill the area behind the weld completely, and the material is pulled onto the surface after mixing, resulting in an uneven weld surface and cavity formation. This lack of filling manifests itself on the weld surface.

The average values from the results of the tensile, impact, and hardness tests of the FS-welded PP/NBR plates using the specified welding parameters are presented in Figure 3, Figure 4 and Figure 5. By increasing the rotational speed, the amount of shear stress applied to the material is increased, leading to better dispersion of NBR in the base material. At low traverse speeds, the tensile strength is increased due to the sufficient time given to mixing the rubber phase into the base material (see Figure 3). In contrast, at a rotational speed of 560 rpm and high traverse speeds (40 mm/min), the tensile strength falls from 26.3 MPa to 13.3 MPa (around 50%), which could be related to the lack of sufficient heat generation.

As the rotational speed increases, there is a clear increase in impact strength. Therefore, by increasing the rotation speed, the distribution of NBR in the matrix becomes far more uniform. However, at higher traverse speeds, there is a slight decline in impact strength due to the reduction of the mixing time, poor distribution of the NBR particles, and incomplete mixing of the polymeric chains (see Figure 4). As can be seen in Figure 4b, the FSW process caused finer rubber particles (3.9 µ) in 20 wt.% NBR loaded PP/NBR samples compared to the twin screw extrusion method (2.4 µ), as calculated by ImageJ software (Laboratory for Optical and Computational Instrumentation, Madison, WI, USA). This was due to the action of stirring and tool movement during the FSW process. The higher impact strength of the FS-welded samples (44.8 j/m) over twin screw extruded samples (44.3 j/m) may be due to the uniform distribution of NBR droplets, together with decreased rubber size as a result of proper process parameter selection.

As the rotational speed increased, the hardness of PP/NBR first reached a peak of about 47 (SD) at 900 rpm, then declined sharply. At a rotational speed of 1120 rpm and higher traverse speed, the hardness of the PP/NBR blend decreased significantly due to the better distribution of NBR (see Figure 5). The rubber phase in the blend played a significant role in this reduction. At a rotational speed of 900 rpm, the higher traverse speed had the opposite effect on the hardness of the PP/NBR blend. Owing to the softening effect of elastomeric phases, the addition of NBR resulted in reduced hardness. Furthermore, after the FS welding procedure, it was observed that the mechanical properties had deteriorated in comparison to the extruded PP/NBR composite plates. The optimum mechanical properties relative to the base material were achieved at a rotational speed of 1120 rpm and a traverse speed of 40 mm/min.

One of the drawbacks of polypropylene is its relatively low impact resistance. Joining the material may lead to the deterioration of its mechanical properties, including impact strength. By adding an NBR phase to a PP matrix, this disadvantage could be eliminated. However, adding NBR to the PP matrix reduces the tensile strength. Mechanical tests revealed a 21.7% reduction in hardness and a 5.4% reduction in the tensile strength of joints after joining extruded samples of PP/NBR (80/20) via heat-assisted FSW using a threaded straight cylindrical pin profile at a rotational speed of 1120 rpm and traverse speed of 40 mm/min. There was also no significant change in the impact strength of extruded PP/NBR composite materials compared to FS-welded composites. To address the reduction of tensile strength and hardness, different fractions of HNTs were introduced (3, 5, and 7 wt.%) into the PP/NBR blend in the presence of a dual compatibilizer, SEBS-g-MA and PP-g-MA. As can be seen from Table 3, the compatibilizers significantly improved the tensile strength of the HNT-filled PP/NBR blends at a weight ratio of 80/20 (wt/wt) compared to the blends without compatibilizers. The enhanced compatibility between rubber and HNT contributed to an increase in the tensile strength of the PP/NBR/HNT nanocomposites. The increased polarity of the rubber and compatibility with the HNT was achieved by the formation of interactions between the succinic anhydride groups grafted onto the NBR molecules and the hydroxyl groups on the surfaces of the HNT. A possible schema for these interactions is shown in Figure 6. Recent research has focused on similar interactions that take place between the hydroxyl groups of HNTs and maleic anhydride groups [43].

Due to the smaller droplet size of the NBR phase, a considerable increment in tensile strength was obtained for blends filled with 7 wt.% HNTs in the presence of PP-g-MA and SEBS-g-MA. Using compatibilizers in the HNT-filled PP/NBR blends improved the impact strength substantially compared to the blends in which no compatibilizers were used. The maximum impact strength was achieved with 3 wt.% of HNTs, both with and without PP-g-MA and SEBS-g-MA, and declined at higher concentrations. However, the improvement in impact strength in the presence of PP-g-MA and SEBS-g-MA for blends filled with 3 wt.% HNTs was considerable. This could be attributed to the refinement in the droplet diameter of the dispersed NBR phase. The presence of PP-g-MA and SEBS-g-MA also increased the impact strength of blends filled with 3 wt.% HNTs due to better dispersion of HNTs in the matrix. At higher concentrations, the impact strength was reduced owing to the aggregation of dispersed HNTs in the PP/NBR blend matrix. Despite the fact that HNTs with fractions up to 5 wt.% improved impact strength, a higher ratio of 7 wt.% HNTs resulted in an 11.6% reduction in impact strength. By incorporating the proper content of HNTs and rubber phase into the PP matrix, a suitable balance of mechanical characteristics can be achieved, allowing for the efficient production and joining of large polymeric structures.

Figure 7 shows the EDAX micrographs of impact-fractured specimens of PP/NBR blends filled with 3, 5, and 7 wt.% HNTs. Figure 7b shows the sample containing 5 wt.% of HNTs. The HNTs are well dispersed. However, in the sample containing 7 wt.% HNTs shown in Figure 7c, there is evidence of an accumulation of HNTs. The decrease in impact strength may therefore be related to the fact that the HNTs formed stress concentration points [44].

Since both the addition of the NBR phase and the FSW process caused a reduction in weld strength, the FS welding procedure was performed with the optimum welding parameters on PP/NBR samples with different contents of HNTs (3, 5, and 7 wt.%). As can be seen in Table 3, there were no large changes in the tensile strength and hardness values obtained compared to the extruded samples. In contrast, there is a noticeable increase in impact strength values for all FS-welded samples. This is because the welding process resulted in a much more uniform distribution of HNTs in the PP/NBR blends. Figure 8a–c shows SEM-EDAX micrographs of FS-welded PP/NBR/HNT nanocomposite samples with different fractions HNTs. As can be seen, the best result was achieved by the PP/NBR composite containing 7 wt.% HNTs (Figure 8c). Figure 8d shows two silicate nanotubes that are entirely encased in the matrix. This suggests that the nanotubes’ surfaces interacted with the matrix, creating a filler-matrix contact. These interactions may lead to the formation of the crack-bridging halloysite-matrix seen in the figure. These unique failure processes are responsible for the enhanced ductility of the FS-welded nanocomposite materials.

### Confirmation Tests

To determine the optimal pin profile, the PP/NBR/HNT nanocomposite was welded using three different pin profiles: a threaded cylindrical pin, a threaded square pin, and a threaded conical pin. The rotational speed was 1120 rpm, and the traverse speed was 40 mm/min. As can be seen in Table 4, FS welding with a threaded cylindrical pin produced better results than welding with the other two pins. This was due to the larger contact surface and disturbance created in the process area.

## 4. Conclusions

In this study, PP/NBR/HNT nanocomposite materials were successfully manufactured and joined via twin screw extrusion or FSW processes. Dual compatibilizers (PP-g-MA and SEBS-g-MA) were applied to improve the mechanical properties of the PP matrix nanocomposites. The microstructural and mechanical properties of the samples were evaluated. Better dispersion of particles was observed for the nanocomposites containing 20% NBR compared to 70/30 (wt/wt) PP/NBR composites. Although increased NBR loading in conjunction with the FSW procedure reduced the nanocomposite tensile strength and hardness, by adding HNTs and dual compatibilizers, it was possible to achieve acceptable mechanical properties for the entire range of the investigated materials. By introducing 7 wt.% of HNTs in optimally FS-welded samples, a noticeable improvement in mechanical characteristics was achieved. This is attributable to the stirring movement of the FSW tool, which reduced the size of the NBR droplets and led to a more uniform distribution of the HNTs. Overall, the choice of materials and welding parameters had a considerable impact on the PP/NBR/HNT nanocomposite materials. The improved mechanical characteristics of FS-welded and extruded PP/NBR/HNT nanocomposites can be attributed to the reduced average rubber droplet size and uniform distribution of HNTs.

## Figures and Tables

**Figure 1 nanomaterials-13-00049-f001:**
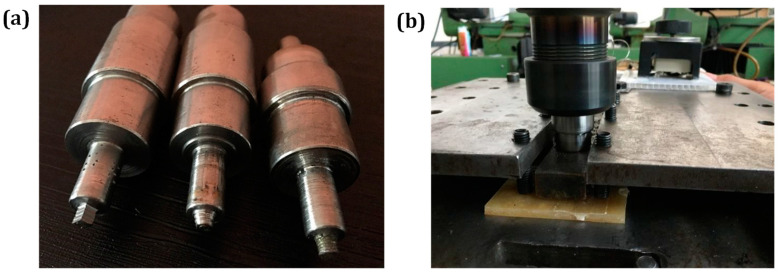
Photographs of (**a**) selected pin profiles (**b**) FSW process.

**Figure 2 nanomaterials-13-00049-f002:**
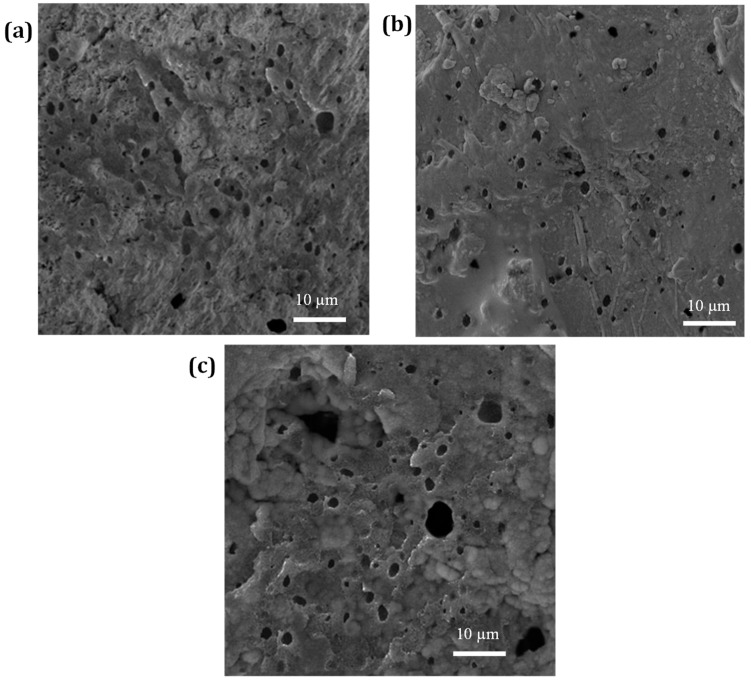
SEM micrographs of etched PP/NBR blends with (**a**) 10 wt.% NBR (**b**) 20 wt.% NBR (**c**) 30 wt.% NBR.

**Figure 3 nanomaterials-13-00049-f003:**
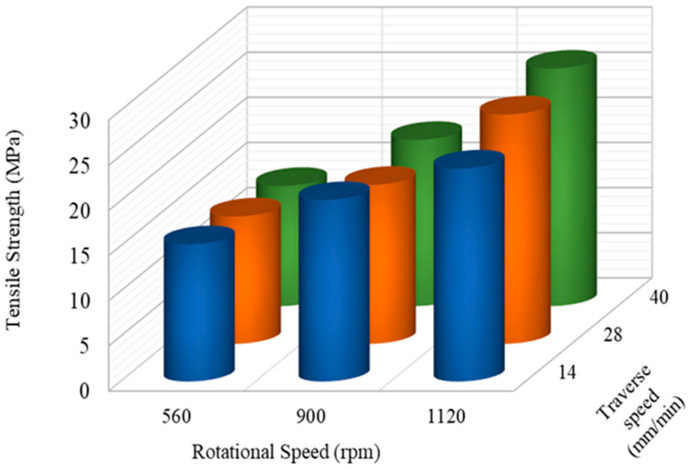
Influence of process parameters on the tensile strength of joints.

**Figure 4 nanomaterials-13-00049-f004:**
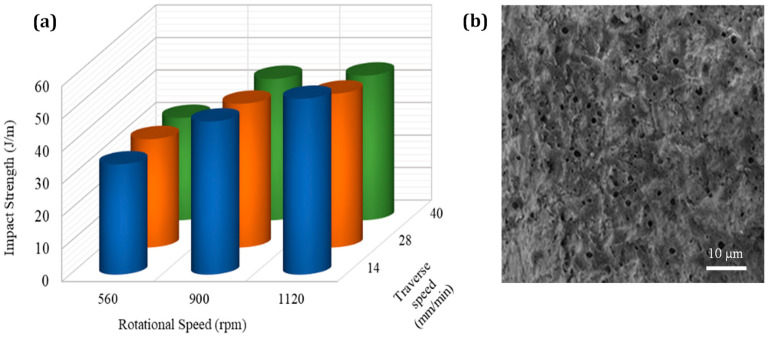
Influence of process parameters on (**a**) impact strength and (**b**) microstructure of joints.

**Figure 5 nanomaterials-13-00049-f005:**
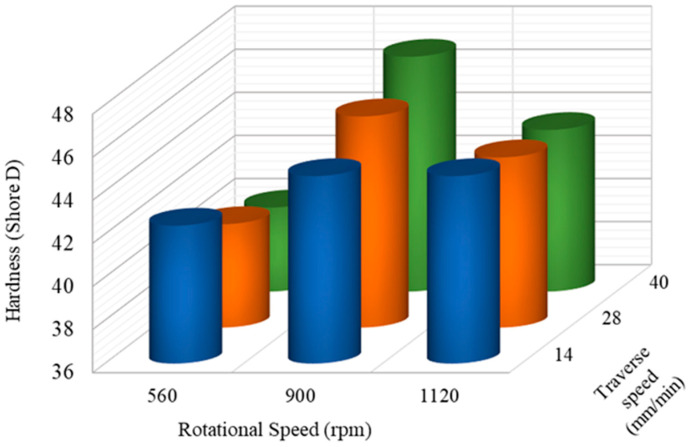
Influence of process parameters on the hardness of joints.

**Figure 6 nanomaterials-13-00049-f006:**
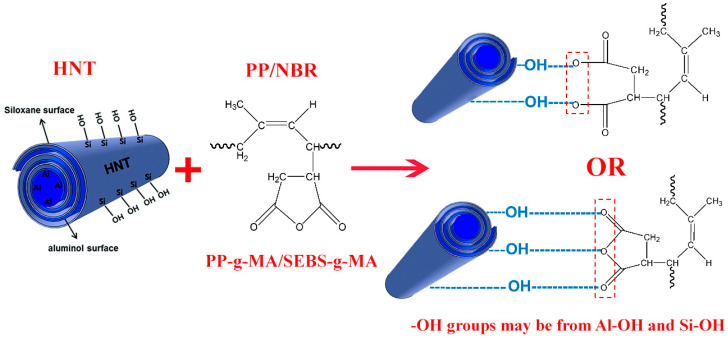
Possible interactions of PP/NBR Blend with PP-g-MA/SEBS-g-MA and HNTs.

**Figure 7 nanomaterials-13-00049-f007:**
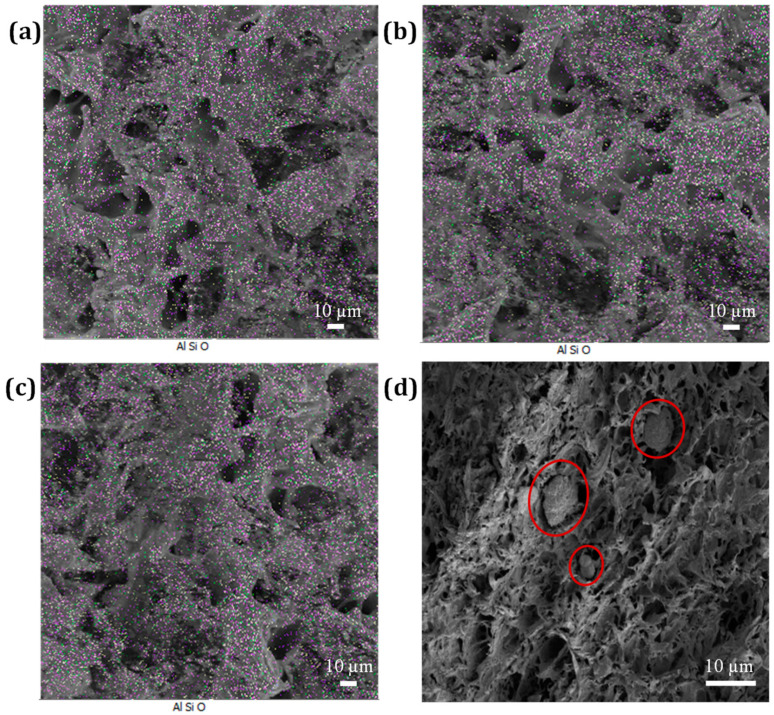
EDX maps of impact-fractured specimens of PP/NBR blends filled with different fractions of (**a**) 3 wt.%, (**b**) 5 wt.%, and (**c**) 7 wt.% HNTs and (**d**) agglomerate (7 wt.% HNTs).

**Figure 8 nanomaterials-13-00049-f008:**
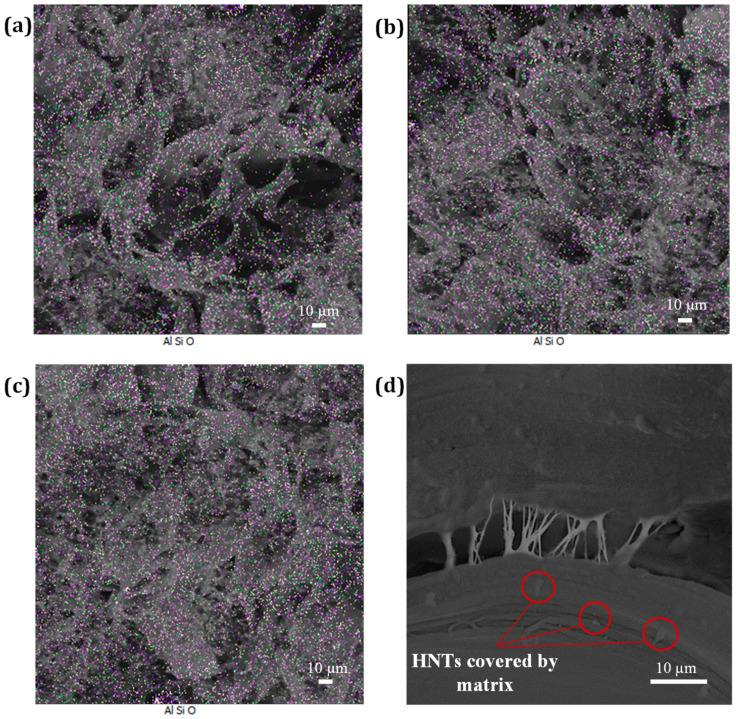
SEM-EDAX micrographs of FS-welded PP/NBR/HNT nanocomposite samples with (**a**) 3 wt.%, (**b**) 5 wt.%, and (**c**,**d**) 7 wt.% HNTs.

**Table 1 nanomaterials-13-00049-t001:** Input parameters and values of each factor.

Input Parameters	Unit	Symbol	Values
Tool rotational speed	rpm	N	560–900–1120
Tool traverse speed	mm/min	V	14–28–40
HNT content	wt.%	H	3–5–7

**Table 2 nanomaterials-13-00049-t002:** Nomenclature and mechanical characteristics of PP/NBR blends.

Sample Code	Process	Composition				Tensile Strength (MPa)	Impact Strength (J/m)	Hardness (SD)
		PP	NBR	PP-g-MA (wt.%)	SEBS-g-MA (wt.%)			
EPNPS8	Extrusion	82	8	5	5	30.1 ± 0.2	37.7 ± 0.1	55.6 ± 0.6
EPNPS18	Extrusion	72	18	5	5	27.8 ± 0.1	44.3 ± 0.3	55.6 ± 0.6
EPNPS28	Extrusion	62	28	5	5	26.2 ± 0.1	47.4 ± 0.2	55.6 ± 0.6

**Table 3 nanomaterials-13-00049-t003:** Comparison of mechanical properties of extruded and FS-welded PP/NBR blends with or without dual compatibilizer and HNTs.

Sample Code	Process	Composition					Tensile Strength (MPa)	Impact Strength (J/m)	Hardness (SD)
		PP	NBR	HNT (wt.%)	PP-g-MA (wt.%)	SEBS-g-MA (wt.%)			
FPNPS18	FSW	72	18	0	5	5	26.3 ± 0.2	44.8 ± 0.3	46.9 ± 0.3
EPNH3	Extrusion	77.5	19.5	3	0	0	22.5 ± 0.1	33.8 ± 0.1	58.9 ± 0.5
EPNH5	Extrusion	76	19	5	0	0	27.3 ± 0.3	32.3 ± 0.3	63.7 ± 0.4
EPNH7	Extrusion	74.5	18.5	7	0	0	30.3 ± 0.2	29.2 ± 0.1	68.3 ± 0.6
EPNHPS3	Extrusion	69.6	17.4	3	5	5	28.1 ± 0.1	49.7 ± 0.2	62 ± 0.1
EPNHPS5	Extrusion	68	17	5	5	5	29.4 ± 0.1	45.6 ± 0.2	70 ± 0.2
EPNHPS7	Extrusion	66.4	16.6	7	5	5	32.1 ± 0.2	39.4 ± 0.2	75 ± 0.3
FPNHPS3	FSW	69.6	17.4	3	5	5	27.4 ± 0.3	49.9 ± 0.3	62.7 ± 0.4
FPNHPS5	FSW	68	17	5	5	5	28.5 ± 0.3	53.6 ± 0.2	71 ± 0.3
FPNHPS7	FSW	66.4	16.6	7	5	5	32.3 ± 0.2	58.2 ± 0.3	76.4 ± 0.4

**Table 4 nanomaterials-13-00049-t004:** Confirmation experiments using different pin profiles.

Material	FSW Tool Pin Profile	Tensile Strength (MPa)	Impact Strength (J/m)	Hardness (SD)
7% HNTs-loaded PP/NBR	Threaded cylindrical	32.1 ± 0.3	57.9 ± 0.4	76.2 ± 0.5
7% HNTs-loaded PP/NBR	Threaded square	28.5 ± 0.1	51.5 ± 0.2	67.8 ± 0.2
7% HNTs-loaded PP/NBR	Threaded conical	29.2 ± 0.3	53.6 ± 0.1	69.6 ± 0.3

## Data Availability

Data are contained within the article.

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
