# Peer review of "Manufacturing and Joining PP/NBR Blends in the Presence of Dual Compatibilizer and Halloysite Nanotubes"

_nanomaterials, 2022, doi:10.3390/nano13010049_

Round 1

Reviewer 1 Report

The authors should use an English language editing service. See my edited copy of the paper that is attached.

The conclusion should not be written in bulleted form. A narrative is the required format.

No discussion, all engineering and science manuscripts have a discussion. It is a self-critique of their work, a comparison with similar methods, with amplification of what was achieved.

Author Response

Response to the queries/comments/suggestions of the reviewers

Queries/criticism/suggestions are presented in order of the list of comments in blue Arial font. Responses follow in black Times Roman. The revised text reproduced from the manuscript is presented in dark red Times Roman.

Detailed responses to Reviewer 1

  1. The authors should use an English language editing service. See my edited copy of the paper that is attached;

Thanks for a very good suggestion. The paper is checked again and recommended parts are edited to improve the language of the manuscript. We have done thorough English editing.

  1. The conclusion should not be written in bulleted form. A narrative is a required format;

Thanks for your recommendation; now the conclusion section includes: (Page 11, Lines 320-336)

“In this study, PP/NBR/HNT nanocomposite materials were successfully manufactured and joined via twin screw extrusion or FSW processes. Dual compatibilizers (PP-g-MA and SEBS-g-MA) were applied to improve the mechanical properties of the PP matrix nanocomposites. The microstructural and mechanical properties of the samples were evaluated. Better dispersion of particles was observed for the nanocomposites containing 20% NBR compared to 70/30 (wt/wt) PP/NBR composites. Although increased NBR loading in conjunction with the FSW procedure reduced the nanocomposite tensile strength and hardness, by adding HNTs and dual compatibilizers it was possible to achieve acceptable mechanical properties for the entire range of the investigated materials. By introducing 7 wt.% of HNTs in optimally FS welded samples, a noticeable improvement in mechanical characteristics was achieved. This is attributable to the stir-ring movement of the FSW tool, which reduced the size of the NBR droplets and led to a more uniform distribution of the HNTs. Overall, the choice of materials and welding parameters had a considerable impact on the PP/NBR/HNT nanocomposite materials. The improved mechanical characteristics of FS welded and extruded PP/NBR/HNT nanocomposites can be attributed to the reduced average rubber droplet size and uniform distribution of HNTs.”

  1. No discussion, all engineering and science manuscripts have a discussion. It is a self-critique of their work, a comparison with similar methods, with amplification of what was achieved;

We appreciate the reviewer's observation; in fact, the discussion section has already been included as the Results and Discussion section and the authors followed the theme of the nanomaterials journal. Now the discussion is also added to this section and necessary parts have been added.

Reviewer 2 Report

This manuscript reports the blending of halloysite nanotubes with poly(propylene) in the presence of two compatibilizers. The resulting blends are referred to as composites although the mechanical properties of the blended material are not always greater than that of the matrix alone. A brief justification for the use of this terminology is probably appropriate.

The manuscript will need significant revision for accuracy, clarity and readability. Corrections are penciled-in directly on pages of the manuscript attached. These are illustrative of the kinds of changes needed throughout. In rewriting, careful attention should be paid to the use of articles, tenses and proper sentence structure. Author's names, et.al. and personal pronouns should be omitted. Superfluous phrases such as "in this study" and "the authors presented" should be avoided.

This manuscript can be substantially improved by some careful editing.

Author Response

Response to the queries/comments/suggestions of the reviewers

Queries/criticism/suggestions are presented in order of the list of comments in blue Arial font. Responses follow in black Times Roman. The revised text reproduced from the manuscript is presented in dark red Times Roman.

Detailed responses to Reviewer 2

  1. This manuscript reports the blending of halloysite nanotubes with poly(propylene) in the presence of two compatibilizers. The resulting blends are referred to as composites although the mechanical properties of the blended material are not always greater than that of the matrix alone. A brief justification for the use of this terminology is probably appropriate;

Thanks to the reviewer for noticing; the Discussion section is now updated as: (Pages 7-8, Lines 241-277)

“One of the drawbacks of polypropylene is its relatively low impact resistance. Joining the material may lead to the deterioration of its mechanical properties, including impact strength. By adding an NBR phase to a PP matrix, this disadvantage could be eliminated. However, adding NBR to the PP matrix reduces the tensile strength. Mechanical tests revealed a 21.7% reduction in hardness and a 5.4% reduction in the tensile strength of joints after joining extruded samples of PP/NBR (80/20) via heat-assisted FSW using a threaded straight cylindrical pin profile at a rotational speed of 1120 rpm and traverse speed of 40 mm/min. There was also no significant change in the impact strength of extruded PP/NBR composite materials compared to FS welded composites. To address the reduction of tensile strength and hardness, different fractions of HNTs were introduced (3, 5, and 7 wt.%) into the PP/NBR blend in the presence of a dual compatibilizer, SEBS-g-MA, and PP-g-MA. As can be seen from Table 3, the compatibilizers significantly improved the tensile strength of the HNT-filled PP/NBR blends at a weight ratio of 80/20 (wt/wt) compared to the blends without compatibilizers. The enhanced compatibility between the rubber and HNT contributed to increasing the tensile strength of the PP/NBR/HNT nanocomposites. The increased polarity of the rubber and compatibility with the HNT was achieved by the formation of interactions between the succinic anhydride groups grafted onto the NBR molecules and the hydroxyl groups on the surfaces of the HNT. A possible schema for these interactions is shown in Figure 6. Recent research has focused on similar interactions that take place between the hydroxyl groups of HNTs and maleic anhydride groups [40].

Due to the smaller droplet size of the NBR phase, a considerable increment in tensile strength was obtained for blends filled with 7 wt.% HNTs in the presence of PP-g-MA and SEBS-g-MA. Using compatibilizers in the HNT-filled PP/NBR blends improved the impact strength substantially compared to the blends in which no compatibilizers were used. The maximum impact strength was achieved with 3 wt.% of HNTs, both with and without PP-g-MA and SEBS-g-MA, and declined at higher concentrations. However, the improvement in impact strength in the presence of PP-g-MA and SEBS-g-MA for blends filled with 3 wt.% HNTs was considerable. This could be attributed to the refinement in the droplet diameter of the dispersed NBR phase. The presence of PP-g-MA and SEBS-g-MA also increased the impact strength of blends filled with 3 wt.% HNTs due to better dispersion of HNTs in the matrix. At higher concentrations, the impact strength was reduced owing to the aggregation of dispersed HNTs in the PP/NBR blend matrix. Despite the fact that HNTs with fractions up to 5 wt.% improved impact strength, a higher ratio of 7 wt.% HNTs resulted in an 11.6% reduction in impact strength. By incorporating the proper content of HNTs and rubber phase into the PP matrix, a good balance of mechanical characteristics can be achieved, allowing for the efficient production and joining of large polymeric structures.”

  1. The manuscript will need significant revision for accuracy, clarity and readability. Corrections are penciled in directly on the pages of the manuscript attached. These are illustrative of the kinds of changes needed throughout. In rewriting, careful attention should be paid to the use of articles, tenses and proper sentence structure. Author's names, et.al. and personal pronouns should be omitted. Superfluous phrases such as "in this study" and "the authors presented" should be avoided;

Thanks to the dear reviewer’s careful attention. We have improved our manuscript by rewriting mentioned sections and possible corrections have been made.

  1. This manuscript can be substantially improved by some careful editing;

Thanks for your kind comments and good suggestions; Done.

Reviewer 3 Report

The paper "Manufacturing and Joining of PP/NBR Blends in the Presence of Dual Compatibilizer and Halloysite Nanotubes (HNTs)" presents interesting results and can be published after major revision. Some issues should be clarified. 

Please delete the acronym HNTs from the title. Please add full decoding of abbreviations to the abstract.

In work were used two different types of compatibilizers (PP-g-MA and SEBS-g-MA) but it isn't easy to understand the feasibility of this approach. What result will be achieved by using one of the compatibilizers or if any compatibilizer will not be used? Appropriate discussion should be added. 

Information about other types of compatibilizers that are effective for similar systems should be added in the introduction.

The scheme which illustrates interactions PP/NBR Blend with PP-g-MA/SEBS-g-MA and HNTs should be added to the paper.

Please cite relevant papers where similar approaches to improve the compatibility of polypropylene-based systems were proposed. 

https://doi.org/10.1002/masy.200450638

https://doi.org/10.1016/j.reactfunctpolym.2010.11.028

Round 2

Reviewer 2 Report

This manuscript is significantly improved.

Reviewer 3 Report

The authors have answered all of my comments and the paper can be accepted for publication in its present form.